# The Hepatoprotective Effects of *Camellia sinensis* on Cisplatin-Induced Acute Liver Injury

**DOI:** 10.3390/life14091077

**Published:** 2024-08-28

**Authors:** Adnan Yilmaz, Fatih Dizman, Kerimali Akyildiz, Sibel Mataraci Karakas, Tolga Mercantepe, Huseyin Avni Uydu, Levent Tumkaya, Koksal Ozturk

**Affiliations:** 1Department of Biochemistry, Faculty of Medicine, Recep Tayyip Erdogan University, 53010 Rize, Turkey; kerimali.akyildiz@erdogan.edu.tr (K.A.); sibel.karakas@erdogan.edu.tr (S.M.K.); 2Recep Tayyip Erdogan University Research and Training Hospital, 53010 Rize, Turkey; fatihdizman0653@gmail.com (F.D.); koksal6161.5@gmail.com (K.O.); 3Department of Histology and Embryology, Faculty of Medicine, Recep Tayyip Erdogan University, 53010 Rize, Turkey; tolga.mercantepe@erdogan.edu.tr; 4Department of Biochemistry, Faculty of Medicine, Samsun University, 55000 Samsun, Turkey; huseyinavni.uydu@samsun.edu.tr; 5Department of Histology and Embryology, Faculty of Medicine, Ondokuz Mayis University, 55000 Samsun, Turkey; levent.tumkaya@erdogan.edu.tr

**Keywords:** cisplatin, white tea, liver, oxidative stress, rat

## Abstract

Acute liver injury is an increasing global health problem. It is a widespread side effect of cisplatin treatment in the clinic and can lead to liver failure if not treated promptly. Previous studies have revealed that green tea can protect some organs from treatments. However, the potential of white tea to prevent the negative effects of acute liver injury has not been addressed so far. The purpose of this study was to investigate the reduction in cisplatin-induced liver injury in rats receiving white tea. Female Sprague Dawley rats with similar weight were selected in this study. Twenty-four rats were divided into three groups of eight animals each and ad libitum nutrition was provided. The control and cisplatin groups were given tap water only, while the white tea + cisplatin group received white tea at a 0.5% weight/volume concentration for four weeks. At the end of the fourth week, the white tea + cisplatin group and the cisplatin group received a single dose of cisplatin (7 mg/kg) via the intraperitoneal route. Five days after that procedure, the rats were anesthetized. Liver tissues and blood samples were collected, which were used for biochemical and histopathological analyses. According to biochemical results, liver tissue MDA and GSH, serum ALT, and AST levels significantly increased in the cisplatin group compared to the control group. Compared with the cisplatin group, although MDA, AST, ALT, and GSH levels were lower in the white tea + cisplatin group, only GSH levels were statistically different. The examination of histopathological and immunohistochemical findings revealed apoptotic cells, vascular congestion, and sinusoidal dilatation in the cisplatin group compared to the control group. This adverse event decreased in the white tea + cisplatin group compared to the cisplatin group. In conclusion, white tea exhibits an ameliorating effect on cisplatin-induced liver injury.

## 1. Introduction

The *Globocan* 2022 report published by the International Agency for Research on Cancer (IARC) stated that 19.96 million new cases had been diagnosed in that year and that cancer was responsible for 9.74 million deaths [1]. There are more than 100 forms of cancer treatment, but chemotherapy, radiotherapy, and surgery, either alone or in combination when necessary, are some of the most frequently employed therapeutic methods. Chemotherapy aims to kill tumor cells through the oral and/or intravenous administration of cytotoxic or antineoplastic chemotherapeutic chemical agents [2]. However, antineoplastic drugs also damage healthy cells in addition to the tumor cells they destroy, either through their direct effect on DNA or by halting cell division [3,4].

Cisplatin (*cis-diamminedichloroplatinum* (*II*)), an anti-tumoral drug, is an organic platinum derivative and the best known and most frequently used chemotherapeutic agent in the treatment of cancer [5,6]. It is employed in the treatment of several cancer types, including those of the bladder, head, neck, lung, ovary, and testis [7,8,9]. However, hepatotoxicity occurs after nephrotoxicity in a dose-dependent manner following cisplatin use [10].

Cisplatin results in the production of hydroxyl, hydrogen peroxide, and superoxide radicals in tissues. These radicals react with various cellular components such as DNA, protein, and membrane lipids [11]. Moreover, cisplatin exposure induces oxidative stress; increases protein oxidation, malondialdehyde (MDA), and 8-hydroxy-2′-deoxyguanosine (8-OHdG) levels; and decreases GSH content, SOD, GPx, CAT, glutathione-S-transferase (GST), and glutathione reductase (GR) levels [12]. MDA levels are currently measured as a biomarker of lipid peroxidation in living tissues [13]. In addition, GSH, which contains the greatest total thiol group, is the most prominent among the antioxidant enzymes that prevent oxidative stress [14].

Tea is produced from the leaves of the *Camellia sinensis* plant, which belongs to the Theaceae family, and is one of the most popular and consumed beverages worldwide [15]. Depending on the processing technique, tea is most commonly classified as green, black, oolong, or white tea (WT) [16]. WT differs from other teas in that it uses only young leaves of the *C. sinensis* plant. WT has important benefits due to its natural antioxidants with biological effects on human health and its high content of catechins and derivatives, as well as other tea components [17].

With its rich catechin and flavonoid content, particularly epigallocatechin gallate (EGCG), *C. sinensis* has been described as one of a thousand promising phytotherapeutic agents [14]. EGCG is the most abundant, with strong antioxidant activity in white tea, and is responsible for the most cancer chemoprevention [18]. Moreover, recent studies on *C. sinensis* have reported that EGCG, EG, and flavonoids in particular prevent lipid accumulation in liver and liver cancer [19,20]. 

In the literature, no research has been found on the effects of WT on liver damage, which is richer than other teas in terms of contents and produced with less processing [20,21,22,23,24]. The purpose of the present study was to investigate the level of liver damage caused by cisplatin used in chemotherapy and to determine whether chronic white tea consumption can reduce or eliminate that injury. Another aim of this study was to describe different perspectives aimed at treatment or prevention for scientists and physicians working in this field through the results obtained. 

## 2. Materials and Methods

Our study is an experimental animal study and is written according to the ARRIVE 3.0 guideline.

### 2.1. Chemicals

Cisplatin was obtained from Faulding Pharmaceuticals PLC (Lemington Spa, Warwickshire, UK), and the other chemicals employed in the study were procured from Sigma (Sigma Chemicals Co., St. Louis, MO, USA).

### 2.2. The Experimental Animals and Study Groups 

For the animal selection in our study, Sprague Dawley rats were selected depending on the type of research, applicability, and possibilities. The sample size of our study was calculated in accordance with the studies of Arifin et al. Twenty-four female Sprague Dawley rats (age: 12–14 weeks; weight: 250–300 g) were obtained from the Recep Tayyip Erdoğan University Experimental Animals Research Center, Turkey. Approval for the study was granted by the Recep Tayyip Erdoğan University Laboratory Animals Local Ethical Committee (no. 2019/12). The rats were housed in transparent polyethylene cages, each containing eight animals, and were allowed ad libitum access to chow and tap water. The rats were placed into artificially controlled rooms at 23 ± 2 °C in a 12 h light/dark cycle and 55 ± 5% humidity one week before the commencement of the study for adaptation. Following the adaptation stage, the rats were divided into three groups, with eight subjects in each group: control, cisplatin (CP), and cisplatin + white tea (CP + WT).

The control and CP groups were given only tap water, while the CP + WT group received 0.5% (*w*/*v*) white tea added to tap water daily for four weeks. Following this, both the CP + WT and CP groups received a single intraperitoneal (i.p.) dose of 7 mg/kg cisplatin. After that procedure, the rats were anesthetized with ketamine-xylazine, and blood and liver tissue samples were collected.

### 2.3. The Procurement and Preparation of White Tea 

The white tea was obtained from a producer (General Directorate of Tea Enterprises, Rize, Turkey). Appropriate amounts of tea leaves were weighed. The tea was added to boiled water (0.5 g/100 mL) and steeped for 3 min [5]. The content of white tea used in the study was determined by high performance liquid chromatography (HPLC) (Table 1). 

### 2.4. Blood and Liver Tissues Collection 

Rat blood samples were collected from the left ventricle under general anesthesia after 12 h of fasting. The samples were placed into tubes with no anticoagulant and left at room temperature for a few minutes, after which they were centrifuged at +4 °C for 5 min at 3500 rpm and stored at −80 °C until the alanine amino transaminase (ALT) and aspartate amino transaminase (AST) analyses. 

The liver tissues were washed three times in cold (4 °C) 0.9% sodium chloride. A pH 7.4 homogenization solution was prepared, consisting of 20 mM 1 L sodium phosphate + 140 mM potassium chloride [18]. Homogenization solution was added to 0.1 g of liver tissue in the amount of 1 mL. Following homogenization at 30 Hertz for 3 min, the homogenates were centrifuged at 800 g for 10 min at 4 °C, and the resulting supernatants were employed in GSH and MDA analyses.

Liver tissues set aside for histopathological examination were placed into 10% formaldehyde. All tissues were stored at −80 °C until analysis. Caspase-3 activities in liver tissues were determined immunohistochemically.

### 2.5. Biochemical Analysis

#### 2.5.1. Determination of Serum ALT and AST Levels

Serum AST and ALT levels were measured on an Abbott Architect^®^c16000 device (Abbott Diagnostics, Abbott Park, IL, USA). 

#### 2.5.2. Determination of Tissue MDA and GSH Levels

The levels of MDA were measured according to the method of Draper and Hadley. MDA, the end product of lipid peroxidation, was measured with the results being expressed as nmol/g tissue [21]. 

The GSH levels were determined by the Ellman method. In the measurement of GSH levels, the free sulfhydryl groups in the homogenate were determined by a spectrophotometric reading of the color formed, and the data were expressed as nmol/g tissue [22].

### 2.6. Histopathological Analysis

The rat liver tissues were quickly trimmed and fixed for 36 h in a 10% formalin solution (Sigma Aldrich, St. Louis, MO, USA). Following routine procedures, the tissues were passed through an increasing alcohol series and rendered transparent before being embedded in paraffin blocks (Merck, Darmstadt, Germany). Sections 4–5 μm in thickness were then obtained from these blocks using a microtome (Leica, RM2125RT, Nußloch, Germany). These were then stained with hematoxylin (Harris hematoxylin, Merck, Germany) and eosin (H&E) (Eosin G, Merck, Germany) and Goldner’s Masson trichrome (Merck, Darmstadt, Germany). After staining, the tissues were examined under a light microscope (Olympus BX51, Olympus Corporation, Tokyo, Japan) and photographed with an Olympus DP71 camera (Olympus Corporation, Tokyo, Japan).

### 2.7. Immunohistochemical Analysis (IHC)

The avidin–biotin peroxidase complex method was employed to determine the immunoreactivity of the caspase-3 primary antibody used to assay apoptotic hepatocytes. Sections 2–3 µm in thickness taken from the paraffin blocks were placed onto positively charged slides and subjected to deparaffinization. They were next incubated with 3% H_2_O_2_ solution to block endogenous peroxidase activity. Following a 20 min application of a blocking solution to prevent background staining, they were incubated with primary antibodies (Caspase-3, Rabbit polyclonal, Abcam, UK) for 60 min. Following the application of the primary antibody, the tissues were incubated with a secondary antibody (Goat Anti-Rabbit IgG H&L (HRP) ab205718, Abcam, UK). Diaminobenzidine chromogen (DAB Chromogen, Abcam, UK) was then dropped onto the tissues, and an image signal was obtained under a light microscope. Counterstaining was applied with Harris hematoxylin (Merck, Darmstadt, Germany), and the tissues were finally covered with an appropriate closing solution.

### 2.8. Statistical Analysis

For data analysis, the Statistical Package for Social Sciences (SPSS; version 18 for Windows, IBM, Chicago, Illinois, USA) was used. The data obtained as a result of the analyses were evaluated to determine whether they conformed to a normal distribution by using the Shapiro–Wilk test, a QQ plot, kurtosis and skewness values, and Levene’s test. The parametric data were presented as mean ± SD. Differences among groups were analyzed by one-way analysis of variance (ANOVA), and post hoc LSD tests were performed. A *p* value of less than 0.05 was regarded to be statistically significant.

The data obtained from the immunohistochemical analyses were calculated as a median and standard deviation (maximum, minimum) for non-parametric data using SPSS 18.00 software, and intergroup comparisons were performed using the Kruskal–Wallis and Tamhane T2 tests. *p* values < 0.05 were regarded as significant.

## 3. Results

### 3.1. Biochemical Results

Hepatic tissue GSH and MDA levels showed a significant difference in the cisplatin group compared to the control group. Moreover, in the cisplatin + white tea group, MDA and GSH levels decreased compared to the cisplatin group (Table 2).

Serum ALT and AST levels significantly decreased in the cisplatin group compared to the control group and decreased in the cisplatin + white tea group compared to the cisplatin group (Table 3).

### 3.2. Histopathological and Immunohistochemical Results

The examination of the sections stained with hematoxylin–eosin (H&E) under light microscopy revealed a normal structure in the Remark cordons and sinusoids consisting of hepatocytes around the central vein, where the liver tissue exhibited a normal structure (Figure 1A–C).

Widespread vascular congestion, sinusoidal dilatations, and numerous necrotic hepatocytes were observed in the section from the CP group (Figure 1A–C).

A decrease in vascular congestion and sinusoidal dilatations was observed in the CP + WT group (Figure 1A–C)

Microscopic images of the control group sections stained with Goldner’s Masson trichrome exhibited a normal structure in the Remark cordons and sinusoids consisting of normal hepatocytes (Figure 2A,B). Diffuse vascular congestion, sinusoidal dilatations, and numerous necrotic hepatocytes were observed in sections from the CP group (Figure 2A,B). A decrease in vascular congestion and sinusoidal dilatations was observed in the CP + WT group (Figure 2A,B).

Light microscopic images of liver tissues stained with the caspase-3 primary antibody from the control group exhibited hepatocytes with a normal appearance (Figure 3A,B), while apoptotic hepatocytes were observed in the CP group (blue arrow) (Figure 3A,B), and the numbers of apoptotic hepatocytes decreased in the CP + WT group (Figure 3A,B).

Caspase-3 positivity evaluation is made in Table 4 and Table 5. The caspase-3 positivity score of 0.00 ± 0.35 in the control group increased to 3.00 ± 0.46 following cisplatin administration (*p* = 0.00; Table 5). The caspase-3 positivity score of 3.00 ± 0.46 in the CP group decreased to 1.00 ± 0.46 in the CP + WT group (*p* = 0.00; Table 5).

## 4. Discussion

Although chemotherapy is a very effective method in the treatment of cancer and one that can be applied irrespective of the type of cancer or the stage of the disease [23], the side effects of the chemotherapeutic agents and resistant cells represent a limiting factor in the administration of anti-cancer chemotherapy [24]. Cisplatin, one of the most frequently employed chemotherapeutic agents, may result in side effects including nephrotoxicity, hepatotoxicity, ototoxicity, neurotoxicity, myelosuppression, nausea, and vomiting [25]. The kidneys function together with the hepatobiliary system in the expulsion of organic platinum cisplatin-derived metabolites from the body, and accumulation mostly occurs in the kidneys [26]. 

Cisplatin nephrotoxicity is one of the most common side effects at the doses used in clinical practice. Hepatotoxicity is less common, although it can also occur in the case of exposure to high-dose cisplatin. Liver toxicity derives from the liver representing the main site of the metabolism of numerous drugs and chemical substances. After the kidneys, most cisplatin accumulation occurs in the liver, and this gives rise to hepatotoxicity with high-dose use. Hepatic necrosis may develop as a result, and apoptotic lesions can be seen in the liver. Studies concerning cisplatin hepatotoxicity are limited, and the mechanism has not yet been fully explained [27,28,29]. 

Drugs are given concurrently with cisplatin for the purpose of reducing or eliminating cisplatin toxicity [30]. Several studies have also investigated the use of food supplements in addition to drugs for reducing this toxicity [31,32,33]. 

Various parameters are used to evaluate the severity of oxidative stress. The measurement of MDA, a marker of lipid peroxidation resulting from the effects of oxidative damage on cell membrane lipids, and GSH, a highly important non-enzymatic antioxidant found in antioxidant systems that protect cells against the adverse effects of oxidative damage, yields some information and predictions concerning liver tissue damage [34,35].

Studies have shown that cisplatin and toxic substances with effects similar to those of cisplatin increase tissue MDA levels, although tissue GSH levels and antioxidant activities are still controversial [29,31,33,36]. Some studies have reported that cisplatin reduces GSH levels and antioxidant enzyme activities [29,37,38,39], while others have reported no change or even an increase [33,40,41,42].

Almost all studies of the oxidative stress caused by cisplatin have reported that it raises tissue MDA levels, and experimental research such as that by Karahan et al. [33], Tarladaçalışır et al. [28], and Greggi et al. [40] has reported positive findings with some food supplements to eliminate this negativity caused by cisplatin. In the present study, liver tissue MDA levels, which increased in parallel with the use of cisplatin, and liver tissue GSH levels, which rose with the stimulation of antioxidant systems in response to this toxic effect, decreased with the therapeutic use of white tea. The increase in GSH levels in the groups receiving cisplatin and the partial decrease in the white tea group suggest that the antioxidant systems are stimulated to eliminate the adverse situation in the cells resulting from the use of cisplatin. 

ALT is highly expressed in hepatocytes, and thus abnormal levels of ALT tend to be more specific to liver damage. Moreover, AST is found in several tissues, such as liver, kidney, and lung. Elevated levels of ALT and AST in the bloodstream are generally considered to be a sign of liver tissue damage [43,44]. 

In our study, rat serum AST and ALT levels increased in the CP group compared to the control group, but decreased in the CP + WT group. The AST and ALT levels were parallel in the rat liver tissue and serum. This supports the idea of the liver damage being caused by cisplatin and the improvement in this damage resulting from white tea. In their rat study on cisplatin, Martins et al. administered a single 10 mg/kg dose of cisplatin and sacrificed the animals after 72 h. The rats’ serum AST and ALT values increased significantly in the cisplatin group compared to the control group. We ascribe the difference between our hepatotoxicity findings and those of Martin et al. to the differences in the cisplatin doses applied and the life spans of the experimental subjects (acute and chronic) [29]. This also shows the vital importance of effective dose limitation in individuals due to receive cisplatin. 

Among the types of tea, EGCG is most commonly found in white tea. It is considered the most bioactive component of tea and is thought to play an important role in preventing liver damage and cancer. There are no studies on white tea and the effects of white tea on liver damage in the literature. Unlike white tea, the literature has a few studies using other teas. In a study with pu-erh tea, there was a significant decrease in the levels of ALT, AST, and MDA; the opposite was true for GSH in the pu-erh tea group contrary to the cisplatin groups [45]. In another study, Amidi et al. investigated the effects of green tea on cisplatin-induced experimental liver function. According to the results, there was a significant difference in ALT and AST levels in the green tea group compared to the cisplatin group [46]. In other studies, analyses of the oxidative stress parameters (MDA, GSH, etc.) and the liver enzyme levels showed more significant differences in the WT groups. The reason for this is that WT has richer content than other teas [47,48,49,50].

The histopathological findings of the present study revealed a decrease in vascular congestion and sinusoidal dilatations in the liver tissue sections from the CP + WT group compared to the cisplatin group, while the immunohistochemical findings revealed a decrease in apoptotic hepatocytes compared to the CP group. When the biochemical results are evaluated together with the histological findings, the decreases in the biochemical analysis findings in the CP + WT group compared to the CP group strengthen the idea that the use of white tea supports antioxidant defense systems. 

In their histological examination of the effects of vitamins E and C on cisplatin hepatotoxicity, Kanter et al. [42] reported that antioxidants with differing characteristics assisted with reducing cisplatin-derived hepatotoxicity. Our histological findings also revealed a decrease in cisplatin-induced apoptotic hepatocytes in the rats that were administered white tea extracts, similar to the therapeutic findings observed with vitamin E and C antioxidants. This also reinforces the idea that the antioxidant properties of white tea can be as valuable as enzymatic or non-enzymatic antioxidants.

Secondary organ damage due to chemotherapy has always attracted increasing interest in the scientific community. Plant extracts and bioactive dietary components play an important role in protecting human health and well-being. In addition, they have the potential to regulate risk factors and manage symptoms in many pathologies, such as various organ toxicities, obesity, respiratory diseases, and metabolic disorders. In this context, white tea is a powerful antioxidant due to its high catechin and flavonoid content, and it might contribute to reducing the number of patients applying to clinics due to the side effects of chemotherapy with its effect of reducing secondary organ toxicity resulting from chemotherapy.

However, despite all these findings, there are several limitations to this research. First of all, this is an animal model study. Although it represents a pilot study concerning the hepatoprotective effect of *C. sinensis*, our study did not include biochemical, histopathological, and immunohistochemical analyses. Oxidative stress was assessed by measuring MDA and GSH levels in biochemical analyses. The evaluation of oxidative stress should be supported by studies addressing the analyses of other oxidant/antioxidant enzymes and/or proteins. Again, apoptosis was assessed using immunohistochemical analyses in our study. Apoptosis should be supported by studies addressing the Tunel method, as well as intracellular calcium and mitochondrial calcium levels. It now needs to be supported by further research considering oxidant/antioxidant molecules and enzymes to elucidate the cellular damage caused by cisplatin.

## 5. Conclusions

In conclusion, we think that drug and nutrition supplements intended to minimize oxidative injury and eliminate cisplatin-derived side effects also need to be very carefully selected. Although we think that white tea has a positive effect on cisplatin-induced liver damage, we also believe that further studies are now needed for a fuller understanding of its antioxidant properties.

## Figures and Tables

**Figure 1 life-14-01077-f001:**
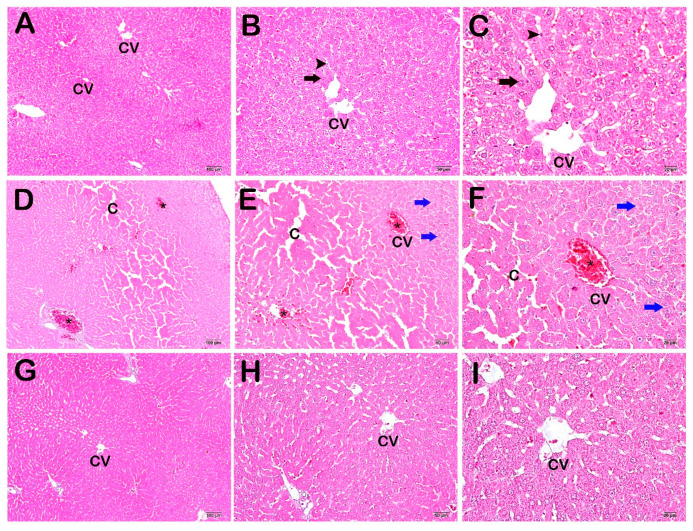
Light microscopic images of liver tissue sections from the control group stained with H&E. (**A**) (×10), (**B**) (×20), (**C**) (×40) Control Group: Normally structured cells are observed in the hepatocytes (arrow) and sinusoids (arrowhead), with a normal central vein (CV) in the healthy liver tissue. (**D**) (×10), (**E**) (×20), (**F**) (×40) CP Group: Diffuse vascular congestion (asterisk) and sinusoidal dilatations (c) observed in the liver parenchyma. In addition, diffuse vacuolization in the necrotic hepatocytes (blue arrow) observed in the hepatic cords. (**G**) (×10), (**H**) (×20), (**I**) (×40) CP + WT Group: A decrease in diffuse vascular congestion and sinusoidal dilatations is observed.

**Figure 2 life-14-01077-f002:**
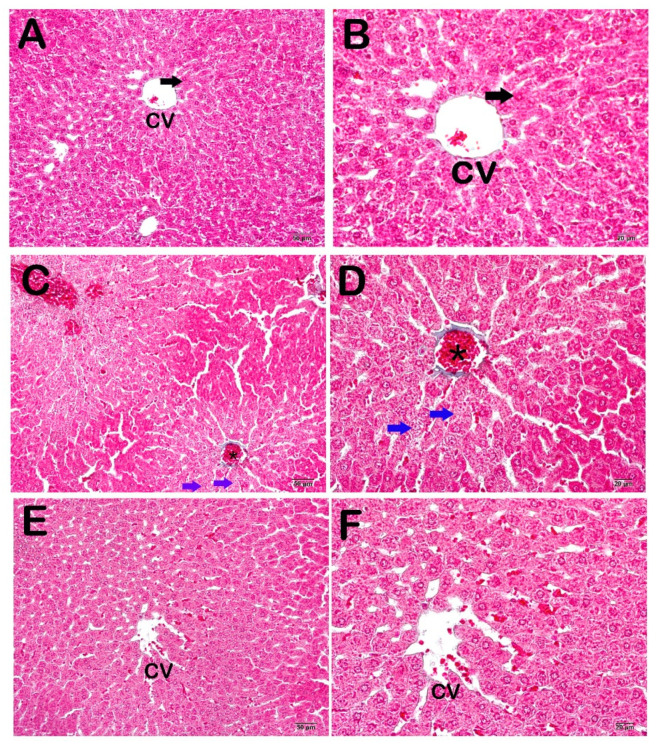
Light microscopic images of liver tissue sections from the control group stained with Masson’s trichrome. (**A**) (×20), (**B**) (×40) Control Group: In the sections of the control group, normal structure hepatocytes (arrow) and central vein (CV) are observed in the hepatic cords. (**C**) (×20), (**D**) (×40) CP Group: The light microscopic examination of sections from the CP group shows a decrease in necrotic hepatocytes (blue arrow) and diffuse vascular congestion (asterisk). (**E**) (×20), (**F**) (×40) CP + WT Group: Light microscopic examination of sections from the CP + WT group shows a decrease in diffuse vascular congestion and sinusoidal dilatations.

**Figure 3 life-14-01077-f003:**
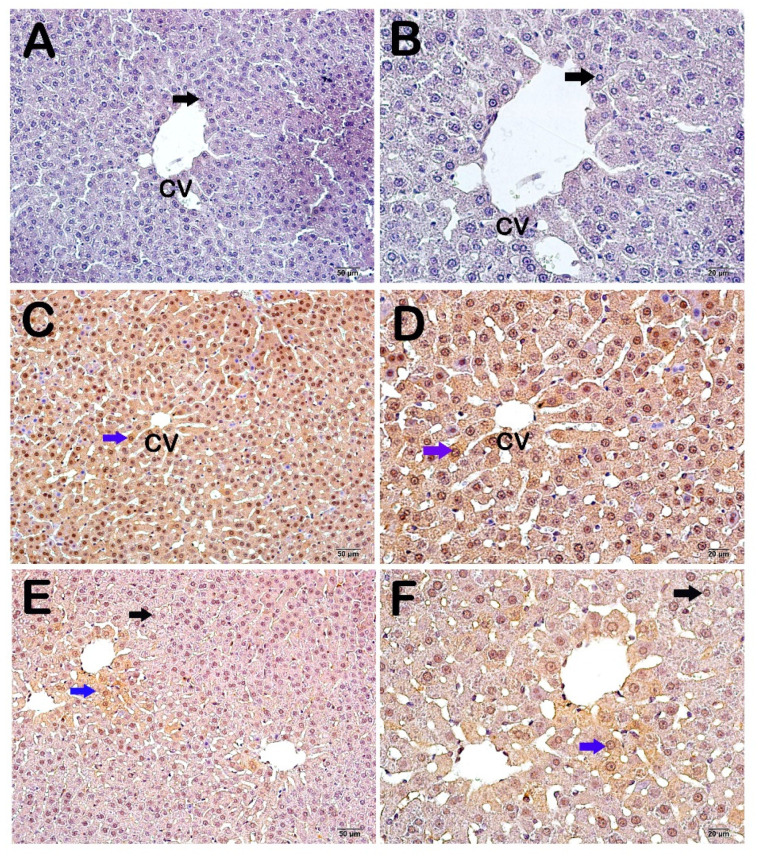
The light microscopic images of the liver tissue sections from the control group stained with the caspase-3 primary antibody. (**A**) (×20), (**B**) (×40) Control Group: It is observed that the normal hepatocytes (black arrow) are immunonegative in terms of Caspase primary antibody (caspase-3 positivity score: 0.00 ± 0.35). (**C**) (×20), (**D**) (×40) CP Group: It is observed that the number of apoptotic hepatocytes (blue arrow) showing intense Caspase-3 immunopositivity is increased (caspase-3 positivity score: 3.00 ± 0.46). (**E**) (×20), (**F**) (×40) CP + WT: It is observed that the number of hepatocytes showing intense Caspase-3 immunopositivity is decreased (caspase-3 positivity score: 1.00 ± 0.46).

**Table 1 life-14-01077-t001:** The catechin and mineral HPLC analysis results of the white tea used in the study.

Component	Amount in Dry Weight
Gallic acid	0.13 (%)
Caffeine	5.62 (%)
EGC (epigallocatechin)	2.54 (%)
EC (epicatechin)	0.91 (%)
EGCG (epigallocatechin gallate)	10.39 (%)
ECG (epicatechin gallate)	3.58 (%)
Copper	0.08 (ppm)
Iron	0.12 (ppm)
Zinc	0.54 (ppm)
Sodium	2.87 (ppm)
Potassium	283.00 (ppm)
Calcium	6.12 (ppm)
Magnesium	26.88 (ppm)
Aluminum	0.98 (ppm)

**Table 2 life-14-01077-t002:** MDA and GSH levels in rat liver tissues.

	MDA (nmol/g Tissue)	GSH (µmol/g Tissue)
Control	1.17 ± 0.11	14.90 ± 2.04
Cisplatin	1.56 ± 0.26 ^a,^*	16.74 ± 2.00 ^a,^**
Cisplatin + White Tea	1.39 ± 0.39	15.08 ± 1.38 ^b,^*

*: *p* < 0.05; **: *p* < 0.01; ^a^: Significantly different compared with the control group; ^b^: Significantly different compared with the CP group.

**Table 3 life-14-01077-t003:** Serum AST and ALT levels.

	AST (U/L)	ALT (U/L)
Control	71.13 ± 4.49	35.25 ± 4.89
Cisplatin	97.88 ± 7.20 ^a,^**	43.75 ± 7.61 ^a,^*
Cisplatin + White Tea	77.00 ± 6.21	40.38 ± 3.89

*: *p* < 0.05; **: *p* < 0.01; ^a^: Significantly different compared with the control group.

**Table 4 life-14-01077-t004:** Caspase-3 positivity scoring table.

Score	Finding
0	Less than 5% caspase-3 positive hepatocytes
1	Less than 25% caspase-3 positive hepatocytes
2	Less than 50% caspase-3 positive hepatocytes
3	More than 50% caspase-3 positive hepatocytes

**Table 5 life-14-01077-t005:** Semi-quantitative analysis results (median ± standard deviation).

Group	Caspase-3 Positivity Score
Control	0.00 ± 0.35
Cisplatin	3.00 ± 0.46 ^a^
Cisplatin+ White tea	1.00 ± 0.46 ^b^

^a^: *p* = 0.00, compared with the control group; ^b^: *p* = 0.00, compared with the CP group. Kruskal–Wallis, Tamhane’s T2 test.

## Data Availability

All data generated for the manuscript are included in the study and are available upon request.

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
