# Peer review of "The Hepatoprotective Effects of Camellia sinensis on Cisplatin-Induced Acute Liver Injury"

_life, 2024, doi:10.3390/life14091077_

Round 1

Reviewer 1 Report

Comments and Suggestions for Authors

The Authors decided to explore the potential protective role of white tea in the course of the oncological treatment with cisplatin. Achieved results appear to be quite promising. From the formal point of view the paper is well designed and well written. What concerns me, is the presence of several meritorious issues that I would like the Authors to deal with:

  1. The term of Camellia Sinensis should be presented in the manuscript in a more clear way. It should be strictly written that it is the type of tea.

  2. The Authors claim in the introduction that they decided to investigate the beneficial effects of white tea in the situation of the treatment with cisplatin due to its antiinflammatory effects, reflected by acting against the oxidative stress. Nevertheless, they mention here its features obtained in diabetic patients and in redox state (references: 15 and 16). How about former studies concerning white tea and cancer or liver disorders? Do we have any existing data in this field, so far? I think such a paragraph should be included.

  3. Please, add to the references more up-to-date positions, e.g.: doi: 10.1016/j.biopha.2019.108604, doi: 10.1016/j.msec.2018.02.026, doi: 10.2147/IJN.S366532

  4. I would like you to mention some other markers of oxidative stress, except MDA and GHS that could be potentially assessed in liver samples to investigate oxidative stress in circumstances presented in the current study.

Author Response

Comments 1.  The term of Camellia Sinensis should be presented in the manuscript in a more clear way. It should be strictly written that it is the type of tea

Response 1. Thank you for your valuable contributions. In the fourth paragraph of the introduction about the definition and properties of white tea was added.

Comments 2. The Authors claim in the introduction that they decided to investigate the beneficial effects of white tea in the situation of the treatment with cisplatin due to its antiinflammatory effects, reflected by acting against the oxidative stress. Nevertheless, they mention here its features obtained in diabetic patients and in redox state (references: 15 and 16). How about former studies concerning white tea and cancer or liver disorders? Do we have any existing data in this field, so far? I think such a paragraph should be included.

Response 2. Thank you for your valuable contributions. Yes, I rearranged the in the fifth paragraph of the introduction about liver and tea and changed references 15 and 16.

Comments 3. Please, add to the references more up-to-date positions, e.g.: doi: 10.1016/j.biopha.2019.108604, doi: 10.1016/j.msec.2018.02.026, doi: 10.2147/IJN.S366532.

Response 3. Thank you for your valuable contributions. I updated it by adding the mentioned references

Comments 4. I would like you to mention some other markers of oxidative stress, except MDA and GHS that could be potentially assessed in liver samples to investigate oxidative stress in circumstances presented in the current study.

Response 4. Thank you for your valuable contributions. Thank you for your valuable contributions. Apart from MDA and GSH, I mentioned some other oxidative stress markers in the third paragraph of the introduction

Reviewer 2 Report

Comments and Suggestions for Authors

Dear Authors,

Congratulations for this study. It was an interesting read, however, several edits are required before publication can be considered:

1. Short title is not necessary. Please remove it

2. The introduction section must have a separate paragraph at the end that includes the study hypothesis and objectives.

3. Include a PICO statement in the materials and methods.

4. All figures should be cited, described and discussed in text in the results.

5. In the discussion section include a paragraph of clinical utility in medicine of these findings.

6. Revise the references section as you include the numbers twice.

Best regards 

Best regards,

Comments on the Quality of English Language

Minor edits are required. 

Author Response

Comments 1. Short title is not necessary. Please remove it.

Response 1. Thank you for your valuable contributions. I removed the short title.

Comments 2. The introduction section must have a separate paragraph at the end that includes the study hypothesis and objectives.

Response 2. Thank you for your valuable contributions. In the last part of the introduction, study objectives added.

Comment 3. Include a PICO statement in the materials and methods.

Response 3. Thank you for your valuable contributions. In addition, since our study is an experimental animal study, it was written according to the ARRIVE 3.O guide. We checked and revised the materials and methods section as per your advice.

Wan Nor Arifin and Wan Mohd Zahiruddin. Sample Size Calculation in Animal Studies Using Resource Equation Approach. Malays J Med Sci. 2017 Oct; 24(5): 101–105.

Comments 4. All figures should be cited, described and discussed in text in the results.

Response 4. Thank you for your valuable contributions. We checked and cited figures entire manuscript as per your advice.

Comments 5. In the discussion section include a paragraph of clinical utility in medicine of these findings.

Response 5. Thank you for your valuable contributions. We checked and added paragraph of clinical utility in medicine in the discussion section as per your advice.

Comments 6. Revise the references section as you include the numbers twice.

Response 6. Thank you for your valuable contributions. Reference section has been rearranged

Reviewer 3 Report

Comments and Suggestions for Authors

The manuscript presents important findings on the hepatoprotective effects of Camellia sinensis but could benefit from more concise writing, a deeper discussion of limitations, and a stronger emphasis on the clinical implications of the research.

Title needs to be imoroved. The authors should mention cisplatin in the title.

The introduction could be more concise. Some sentences are slightly repetitive, which can dilute the impact of the argument. Additionally, the transition between discussing cancer treatments and the specific focus on cisplatin-induced hepatotoxicity could be smoother.

The aims of the study need to be summarized.

The description of the statistical methods used could be more detailed. Additionally, more information on the selection criteria for the rats and any potential confounding factors would be beneficial.

Figure legends need to be revised and enhanced.

Discussion should be strengthen. 

Comments on the Quality of English Language

Minor editing of English language required.

Author Response

Comments 1. Title needs to be imoroved. The authors should mention cisplatin in the title.

Response 1. Thank you for your valuable contributions. The word of cisplatin was added to the title.

Comments 2. The introduction could be more concise. Some sentences are slightly repetitive, which can dilute the impact of the argument. Additionally, the transition between discussing cancer treatments and the specific focus on cisplatin-induced hepatotoxicity could be smoother.

Response 2. Thank you for your valuable contributions. Necessary edits were made in the introduction section. Cisplatin linked to liver and tea

Comments 3. The aims of the study need to be summarized.

Response 3. Thank you for your valuable contributions. The purpose of the study was summarized.

Comments 4. The description of the statistical methods used could be more detailed. Additionally, more information on the selection criteria for the rats and any potential confounding factors would be beneficial.

Response 4. Thank you for your valuable contributions. The statistical methods used are laid out in full detail and ddded information on how to choose  animals

 Comments 5. Figure legends need to be revised and enhanced.

Response 5. Thank you for your valuable contributions. We checked and revised Figures Legends as per your advice.

Comments 6. Discussion should be strengthen

Response 6. Thank you for your valuable contributions. The discussion has been rearranged and new paragraphs have been added.

Reviewer 4 Report

Comments and Suggestions for Authors

I had the pleasure of reviewing the manuscript "The Hepatoprotective Effects of Camellia Sinensis on Antineoplastic Agent-induced Acute Liver Injury" by Yilmaz et al. The authors aimed to prove the hepatoprotective effects of white tea (Camelia Sinensis) against the damage caused by cisplatin, a frequently used antineoplastic agent. The data presented by the authors are interesting, but the quality of the manuscript must be improved before considering its publication.

The authors must thoroughly check the use of the English language and the editing.

The abstract must be improved. At the beginning, there should be some data on the background to present the importance of the study. Also, the presentation of the methods and the results must be imporved.

The authors must rewrite the background/Introduction inside the manuscript and make the section presenting the aim of the study more clearly written.

The authors must present the methods better, clarifying the presentation of those three groups. Also, the duration of the study must be clearly presented. Please clarify whether spring water, tap water, or drinking water is used.

In the Results section, the authors must present the study results once; they do not need to be in the text and also in the Table with the same data (see data from Table 3). The section with the histology results must be improved. 

The authors must better present the study's limitations, not only those already mentioned. They also must discuss the novelties that their study brings to the literature and how their results may help treat patients using cisplatin.

Regarding the Conclusions section, the authors must limit only to conclusions that arise from the study and not some general ideas.

Overall, the quality of the medical writing must be improved.

Comments on the Quality of English Language

The paper needs corrections that better be made by a native English speaker. 

Author Response

Comments 1. The authors must thoroughly check the use of the English language and the editing.

Response 1. Thank you for your valuable contributions. Language revised

Comments 2. The abstract must be improved. At the beginning, there should be some data on the background to present the importance of the study. Also, the presentation of the methods and the results must be imporved.

Response 2. Thank you for your valuable contributions. The summary has been reorganized and improved. Additionally, the method and results section was edited.

Comments 3. The authors must rewrite the background/Introduction inside the manuscript and make the section presenting the aim of the study more clearly written.

Response 3. Thank you for your valuable contributions. Background of the article - the introduction has been reorganized. The purpose of the study is clearly written.

Comments 4. The authors must present the methods better, clarifying the presentation of those three groups. Also, the duration of the study must be clearly presented. Please clarify whether spring water, tap water, or drinking water is used.

Response 4. Thank you for your valuable contributions. Study duration and groups were clearly stated. The type of water used was specified

Comments 5. In the Results section, the authors must present the study results once; they do not need to be in the text and also in the Table with the same data (see data from Table 3). The section with the histology results must be improved. 

Response 5. Thank you for your valuable contributions. Biochemical results were briefly expressed in the findings section. Histological results reconstructed.

Comments 6. The authors must better present the study's limitations, not only those already mentioned. They also must discuss the novelties that their study brings to the literature and how their results may help treat patients using cisplatin.

Response 6. Thank you for your valuable contributions. The discussion has been edited and new relevant information has been added.

Comments 7. Regarding the Conclusions section, the authors must limit only to conclusions that arise from the study and not some general ideas.

Response 7. Thank you for your valuable contributions. The results section was edited and only the results we obtained from the study were stated.

Comments 8. Overall, the quality of the medical writing must be improved.

Response 8. Thank you for your valuable contributions. Text quality improved

Round 2

Reviewer 1 Report

Comments and Suggestions for Authors

The authors modified the manuscript according to the suggested recommendations. I accept the paper in the present form.

Reviewer 2 Report

Comments and Suggestions for Authors

The revised version is significantly improved. Publication can be considered at this point. 

Comments on the Quality of English Language

Only minor English corrections are required. 

Reviewer 3 Report

Comments and Suggestions for Authors

Accept.

Comments on the Quality of English Language

Minor editing of English language required.

Reviewer 4 Report

Comments and Suggestions for Authors

The authors addressed the comments and recommendations for changes in all aspects of the manuscript. Still, some minor editing may be needed, as the use of abbreviated words may require some correction. Besides these minor issues, I have nothing else to recommend.

Comments on the Quality of English Language

Minor editing is needed, and nothing else regarding the quality of the English language.